# The Construction of the Past: Towards a Theory for Knowing the Past

**Kenneth Thibodeau**

Tim Tam Trail, Evergreen, CO 80439, USA; KThibodeau@Fordham.edu

**Abstract:** This paper presents Constructed Past Theory, an epistemological theory about how we come to know things that happened or existed in the past. The theory is expounded both in text and in a formal model comprising UML class diagrams. The ideas presented here have been developed in a half century of experience as a practitioner in the management of information and automated systems in the US government and as a researcher in several collaborations, notably the four international and multidisciplinary InterPARES projects. This work is part of a broader initiative, providing a conceptual framework for reformulating the concepts and theories of archival science in order to enable a new discipline whose assertions are empirically and, wherever possible, quantitatively testable. The new discipline, called archival engineering, is intended to provide an appropriate, coherent foundation for the development of systems and applications for managing, preserving and providing access to digital information, development which is necessitated by the exponential growth and explosive diversification of data recorded in digital form and the use of digital data in an ever increasing variety of domains. Both the text and model are an initial exposition of the theory that both requires and invites further development.

**Keywords:** archival bond; archival science; class diagram; constructed past theory; intentional domain; pragmatic information theory; sphere of interest

---

## 1. Introduction

The past does not exist and never did. The past is always something that is constructed by thinking, writing or speaking about former times. In this respect, knowledge about the past is not different from knowledge gained from direct sensory experience. Neuroscience tells us that the reality we perceive is not a direct reflection of the external world. Rather, perceptions are constructions produced by the brain's predictions or guesses about the causes of incoming sensory signals [1]. The fundamental difference between knowledge gained from perception and knowledge of the past is not in the processes that produce them, but in their sources. Knowledge of the past comes from vestiges and imprints, where vestiges are persistent objects that survive from former times and imprints are previously generated constructions of the past.

'Constructing the past' may bring to mind the writing of history or biography, but the process can take many different forms: management review, audit, courtroom argument, archaeological analysis, psychotherapy, and others. One might assume that all constructions of the past, like these examples, are retrospective; however, even real-time activities, such as keeping a diary, video surveillance, medical registries, live news coverage, and web crawling, produce vestiges of the past by collecting and keeping certain types of data.

Constructions of the past are necessarily incomplete and slanted. Consider the real-time activities just mentioned. Diary entries are selective. Video cameras have limited scope and finite precision. Medical registries are self-selecting. Live news coverage is typically short and limited to topics of presumed public interest. Web crawlers capture data about only a fraction of websites and do

not follow all hyperlinks. Such limitations only increase with the flow of time as the vestiges of past times erode and disappear and imprints more readily bear the perspectives and biases of the contexts in which they are constructed. In fact, both retrospective and real-time constructions of the past are inevitably colored by the imposition of expectations and present purpose. They are often influenced by the results of prior constructions. Furthermore, they cover only limited time periods; address particular interests; use specific sources and finite amounts of data; and apply distinct methods for organizing, processing and rendering data about the past.

This paper proposes a model of the development of knowledge of the past, called Constructed Past Theory (CPT). The domain of CPT encompasses only the process and materials of construction, not the knowledge that results. CPT does rest on certain assumptions about knowledge, most fundamentally the semiotic proposition that to know is to affirm a proposition, where a proposition is an expression that has an objective meaning that can be believed, doubted, denied or asserted as either true or false. It follows that knowing is a process, not a thing. The persistent counterpart of knowing, what we call knowledge, is either a persistent representation of one or more assertions, such as in a document, or the capability for producing or reproducing assertions. This capability assumes (1) a store or stores of data structured in a way that enables their retrieval and presentation in propositions, and (2) the ability to select, retrieve and process data in order to output propositions.

CPT focuses on processes that construct the past from vestiges left behind, rather than those that collect or record real-time data. Its objective is to develop a framework for the discovery and delivery of vestiges from the past, the evaluation of their appropriateness for the purpose a construction has, and their exploitation in the process of construction. CPT also offers the potential to reveal and offset inappropriate preconceptions or predispositions by illuminating the original contexts in which things happened or existed in the past.

Although a basic goal in developing Constructed Past Theory is to support implementation in automated systems, the theory does not accept the distinction between data and information that is common in information technology, as reflected in assertions such as: "data has no meaning or value because it is without context and interpretation;" and "data are discrete, objective facts or observations, which are unorganized and unprocessed, and do not convey any specific meaning." Such assertions are contrasted to information, which is described as "data that have been processed so that they are meaningful;" or "data that have been interpreted and understood by the recipient." [2].

Data are structured or, more accurately, exist within structures that enable their interpretation. Without a specified interpretative structure, it would not be possible consistently and coherently to perform even the basic database operations of create, read, update and delete. "1734" and "Peter" are marks, not data. 'Datum' is the Latin singular of data. It means a given. The answer to the question of what these two signs are can only be that they are strings of numbers and letters, respectively. That assertion gives next to nothing. To qualify as a datum, either sign must be given within a context that enables its interpretation; for example, if it were asserted that "1734" is the highest altitude above sea level, in meters, in the city of Denver, Colorado or that "Peter" is the first name of a science fiction writer whose last name is Cawdron. The latter would be valid if the first and last names were recorded in appropriate fields in a bibliographic database, but the meaning of "Peter" would be completely different if it was found in a list of the apostles of Jesus Christ. If "1734" were written on an isolated and otherwise blank piece of paper, it would be an uninterpretable signal, not a datum. There would be no way to determine if it referred to altitude, a date, a temporary password or anything else. Construed in this way, a datum is a sign as defined in semiotics, "something that stands to somebody for something else in some respect or capacity" [3] (p. 6).

Data can be located in three types of contexts: (1) where the basis of its interpretation is concomitant and explicit, as in the statement about Peter Cawdron, (2) where the basis of its interpretation is concomitant but implicit, as in a structured database, and (3) where the basis of its interpretation is not immediately concomitant, as in a conversation or extended interchange of messages. In the third context, the basis of interpretation may be either explicit or implicit. This type of context is

termed the context of situation in systemic functional linguistics. Whether and to what extent the basis of interpretation is or needs to be made explicit in this context depends on the extent to which the interlocutors share common assumptions. If they share a broad and deep context of culture, even a disfluency, such as "uh-uh," can be a datum [4].

CPT adopts the view of pragmatic information theory that information is not a persistent object, but a phenomenon, one that produces either a behavior or a change of state in a recipient. The phenomenon is the receipt of a set of signals which have to be recognized as such by the recipient and interpreted in some way in order to produce either a behavior or a change of state [5].

This paper is an introductory essay that presents a conceptualization of the production of knowledge of the past but recognizes the necessity of further evolution. The intention is to stimulate additional thinking along these lines with the ultimate objective of contributing to the articulation of a new discipline, called archival engineering, whose assertions are articulated in a manner that is empirically verifiable and wherever appropriate quantitative [6]. The theory and ideas presented here evolved during 20 years of participation as a researcher and research leader in the multidisciplinary International Research on Permanent Authentic Records in Electronic Systems (InterPARES) collaboration [7]. InterPARES 1 investigated conceptual, theoretical and empirical aspects of electronic record keeping systems that managed the digital equivalents of many types of traditional records. InterPARES 2 focused on records that have no hard copy equivalents, specifically in interactive, experiential and dynamic systems. InterPARES 3 tested the results of the first two phases in numerous case studies. InterPARES 4 addressed theoretical, conceptual, legal, contractual and practical issues relating to records preserved in the cloud. In InterPARES 4, the author led the Preservation as a Service for Trust (PaaST) project, which formulated functional and data requirements for preserving records in the cloud. This context is, in essence, a black box, given that cloud service providers typically do not identify either the technologies they use or when they are changed. This forces the articulation of requirements at a high level of abstraction heavily oriented towards inputs and outputs. It also necessitated the formulation of requirements that enable verification that outputs satisfy the theoretical criteria of archival science and the objectives and policies of the clients engaging cloud services [8]. This project, in particular, confirmed the feasibility and value of a reformulation of archival science in abstract and quantitative terms.

Part 2 of this paper presents an overview of the current state-of-the-art. It is necessarily at a very general level because there is no existing counterpart to CPT. Part 2 identifies the conceptual roots of CPT and describes its relationship to other cognate disciplines.

Part 3 articulates the conceptual framework, introducing and defining basic concepts, and providing insights on elaboration and application of CPT in constructing knowledge of the past. Section 3.1 articulates CPT by introducing, at a high level of abstraction, the basic elements involved in constructing the past, including the contexts in which constructions are initiated and developed, the things and events that are the objects of interest in constructing the past, and the materials used in construction. The theory is set out as text supplemented with class diagrams using the notation of UML 2.5 [9]. The classes should be interpreted as abstract, needing further articulation to support any implementation. To signal that the diagrams express a domain model, they do not implement the UML convention of forming names of classes and attributes without spaces between words and they do not identify operations. Names of classes and properties in the diagrams are also capitalized in the text. Instead of 'operation', CPT uses the more generic term, 'behavior.' Similarly, the diagrams lack specifications that would be needed if the model were intended to guide software development. Features such as visibility, namespace, data type, operation signature, return type, concurrency, etc., are not articulated. Other features such as dependency and multiplicity are included only when essential to understanding the concepts depicted. While UML class diagrams most often represent relationships as associations linking classes, the standard categorizes relationships as classifiers [9] (Section 7.8). CPT adopts this perspective, recognizing that a relationship, in addition to linking two other classes, can have attributes that are proper to itself, such as label, direction, weight and multiplicity.

Part 4 shows how the model elaborated in Part 3 could be implemented in a quantitative and testable manner using graph theory. It discusses how a graph theoretical approach can eliminate ambiguity and add greater precision in archival theory, as well as benefit those engaged in constructing the past using the materials preserved in archives and help archival institutions improve the performance of their functions.

Part 5 synthesizes the concepts set out in Part 3 in an overview that also addresses how the model can be applied in undertaking and evaluating constructions of the past.

## 2. State-of-the-Art

There is no existing theory comparable to Constructed Past Theory. One might expect to find such a theory in the discipline of historiography, but historiography is concerned with the products of constructing the past, the literary compositions that historians have created, and how their philosophies, leanings and "takes" on the world shape their use of data in construing their narratives [10,11]. CPT provides a schema for addressing the concerns of historiography under the rubric of the Intentional Domain, supplemented by its representation of the process of construction and the materials used.

Important elements of CPT are rooted in the discipline called archival science. Archival science is described as both a pure and applied discipline that "investigates the reasons and motives of people for creating, managing, using, preserving or destroying records and archives, and the ways in which their records- and archives-related behavior is determined by the social and historical environment in which they display it" [12]. Archival science fits the definition of science as "a branch of knowledge or study dealing with a body of facts or truths systematically arranged and showing the operation of general laws," [13] but it cannot be said that archival science is "a systematic enterprise that builds and organizes knowledge in the form of testable explanations and predictions" [14]. Reports of quantitative tests and even formulations of testable predictions are absent from the archival literature.

Archival theory has not been articulated in a manner that unambiguously supports empirical testing. Archival science is typically expressed in philosophical and rhetorical terms. Some archival concepts are so broadly defined that it can be challenging to determine what they mean or whether they are applied appropriately in various cases. Some are defined so rigidly that they can be difficult to apply in situations where they seem relevant. Others concepts, even basic terms such as archival fonds and record series, are subject to extensive disagreements among experts [15–19]. The way archival concepts have been expressed does not facilitate translation into automated systems. There are several applications that support the work processes of archival organizations. However, none of them actually address the production of knowledge of the past. Rather, they extend at most only to the delivery of archival materials to users, not supporting the construction of the past from those materials [20–24]. A basic motivation for the research behind this paper is to lay the groundwork for reformulating archival theory with the goal of making it an engineering discipline. A basic rationale for archival engineering, rather than science, is that the primary purpose of archival science is to produce insights, understanding and guidance for the creation, management, keeping and use of records in both personal and corporate contexts. The value of archival science is realized only in practical application [25].

Although it is at a higher level of abstraction, the theory laid out in this paper is intentionally consistent with the Records in Context conceptual model and ontology being developed by the International Council on Archives [26]. The theory for constructing the past also builds on what is arguably the most fundamental idea of archival theory, the archival bond. As articulated by Cencetti in the mid-twentieth century, the archival bond is the relationship among documents that arises from their use in an activity by an agent [27]. This relationship is ontologically prior to, and thus independent of, the keeping of documents as records. For example, when one agent sends a message to another and the second agent responds, there is a relationship between the two messages regardless of whether either agent keeps them as records. An archival bond grows over time to include all documents related by their use by an agent in an activity. The concept of the archival bond is used, with modifications, to describe the contemporaneous relationships that exist among documents used

in an activity in Section 3.2 below. That section also demonstrates how data on methods of creating, keeping and managing records can be used in constructing the past.

While it incorporates and builds on established archival theory, CPT is articulated from a perspective that is 180° opposite to that of archival theory, which start from records and builds upward towards the discovery and delivery of relevant records by persons interested in using them. CPT adopts a perspective of articulating what is involved in constructing knowledge of the past and how it is constructed and builds downwards towards the materials that can be used in the construction.

## 3. Constructed Past Theory (CPT)

### 3.1. Architectural Design

The construction of the past may be described as a process of conceiving an initial Target Past and using it to produce a Constructed Past. These two concepts may be compared to the plans and the actual building in physical architecture. They adapt the distinction in Pierce's theory of signs between immediate object, a sign as understood at some point in a semiotic process, and dynamic object, the understanding of the sign at the end of that process [28]. Analogously, the Target Past is the object of a past construction as it is conceived at the outset and evolves during the process. The process of developing a Constructed Past from a Target Past can be a simple or complex affair, ranging from finding a few data items, such as contact information for an individual, to extensive discovery, extraction, interpretation, analysis and synthesis of data and complex argumentation. Extending the analogy to physical architecture, the initial Target Past is like the original blue prints for a building, but progress in construction is likely to lead to changes in the Target Past, comparable to as-built plans for buildings. The Constructed Past is the final product.

Target Past and Constructed Past are represented as two classes in Figure 1, Framework for Construction of the Past, a UML class diagram. Figure 1 shows the fundamental relationship between these two classes: the Target Past leads to the Constructed Past, which is derived from the Target Past.

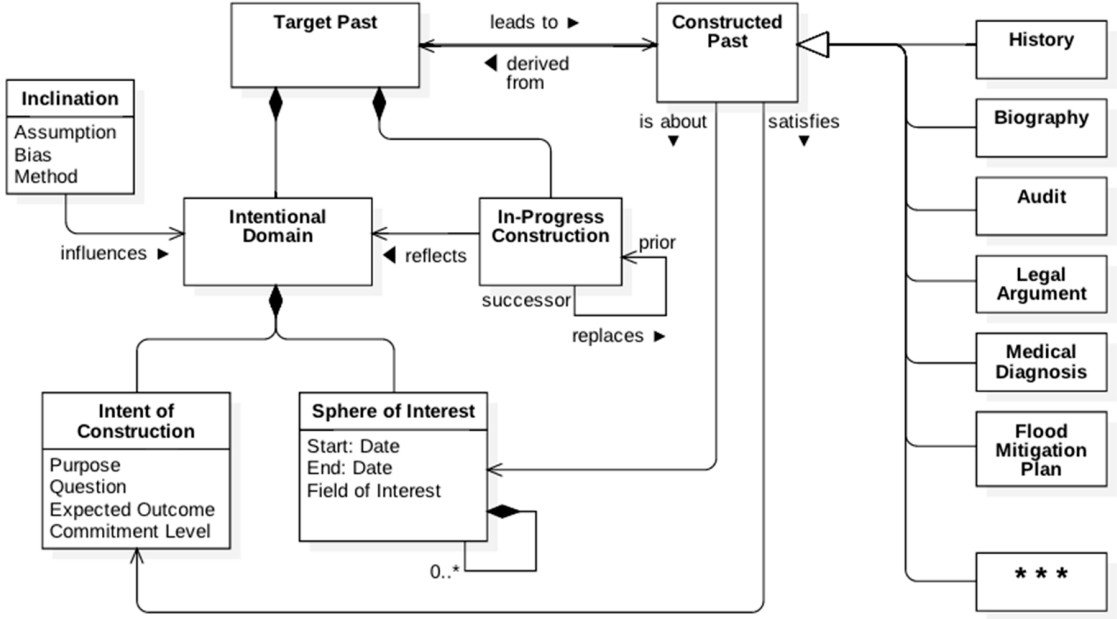

**Figure 1.** Framework for Construction of the Past. The construction of the past centers on the Target Past, which comprises both an Intentional Domain that specifies what is being attempted and why and an In-Progress Construction, which is the state of construction during the process. The final output is a Constructed Past.

The Target Past has two components, Intentional Domain and In-Progress Construction. They are shown related to Target Past by composition; that is, if an instance of the Target Past disappeared, so would its components. The Intentional Domain undergirds and guides the construction. Influenced by the Inclination of the party undertaking the construction, including Assumptions about the past, Bias toward the result of construction and preferred Methods of selecting, processing and presenting data about the Target Past, the Intentional Domain is the foundation on which the past is constructed. The In-Progress Construction encapsulates the results during the process.

Intentional Domain comprises the Intent of Construction and Sphere of Interest. Intent of Construction shapes the process and its results, articulating the Purpose for which the construction is undertaken, identifying the Questions that will be addressed, characterizing the Expected Outcome and indicating the Commitment Level, including the amount of time and effort expected to be expended. Intent of Construction can vary widely. The construction of a Target Past can aim to satisfy personal, professional, communal, or organizational interests. It may be motivated by mere curiosity, deep intellectual interest, or practical concerns, such as safety, material gain, competitive advantage, or other reasons. The Expected Outcome can range across a wide spectrum from minor additions or updates to a database to in depth analyses. The expected outcome can change in the process of construction.

Sphere of Interest specifies the time period under investigation and what is of interest. The time frame can range from a point in time, such as when a specific event happened, to a lengthy span, such as tracing the evolution of a major change in society. Everything within the Sphere of Interest must have existed within its start and end date. A subject of interest may not be bounded by those dates; that is, it may exist previously or afterwards, and it might come into existence after the start date or go out of existence before the end date. However, it must exist at some time within the time frame of the Sphere of Interest. A priori, the things that are of interest in the time frame could be anything that existed or occurred within its limits. In practice, however, the Field of Interest will determine what types of things receive attention. Thus, in a study of military tactics of the U.S. Army in the Indian Wars in the Great Plains, weaponry, and in particular, the introduction of the Colt revolver, will be of interest, while the evolution of property rights in relation to cattle ranching in the same area and period will not [29,30].

At the start of the process, the Sphere of Interest could be explicitly articulated or tacitly assumed. It may be based on extensive awareness of things in the time frame or chosen out of curiosity about something previously unknown. The Field of Interest could be a broad area such as geography, politics, economics, culture, a scientific discipline, social norms, or some combination of these. However, it could also be highly focused, addressing only a single person or event. A Sphere of Interest may contain lower-level Spheres of Interest. Obvious examples are comparative studies, where the same questions are addressed to different, but comparable, empirical domains and multi-disciplinary research, where different types of question are addressed to a single domain. Sphere of Interest is addressed more fully in Section 3.1.1.

The Intent of Construction and the Sphere of Interest are interdependent. Different intents can produce different specifications for what is nominally the same Sphere of Interest. For example, the historic growth of a forest could be investigated for an environmental analysis, for purposes of wildfire management, or to project timber yields. Each would result in different Target and Constructed Pasts [31–33]. Conversely, as the Sphere of Interest is explored, insights gained may lead to modifying the Purpose, causing the Sphere of Interest to be viewed from a different perspective or changing the expectation of what the Constructed Past will be. Things leaned in the process may also lead to modification of the Sphere of Interest, expanding or contracting the time frame or Field of Interest, revealing additional objects or events that should be considered, or reevaluating the importance or relevance of various elements.

At initiation, the Target Past may only reflect a rudimentary notion of what is being investigated. Except in the simplest cases, as the process of construction proceeds, there will be changes in the In-Progress Construction. New elements may be added, others modified or deleted. Thus, Figure 1 includes

a self-referential relationship for In-Progress Construction, with prior versions succeeded by successor ones. The direction of the relationship shown in Figure 1 is from successor to prior because the successor is derived from and replaces the prior version.

The Constructed Past should satisfy the Intent of Construction and be about the Sphere of Interest. The Constructed Past may be knowledge that is kept only in a person's memory, but it could take a tangible and persistent form such as a publication or data in a database. Histories and biographies readily come to mind as examples of persistent Constructed Pasts, but many other types are possible. For example, reports of audits share with histories and biographies that they are extensive descriptions of things in the Sphere of Interest; however, while the purpose of constructing histories and biographies is to produce such descriptions, audits are performed for purposes that go beyond description, namely, assessing conformance with laws, regulations, policies, standards, etc. Other forms of Constructed Past do not aim to describe anything in the past so much as to use data about the past to support present or future-oriented purposes. Examples of such products shown in Figure 1 include Legal Argument, Medical Diagnosis, and Flood Mitigation Plan. The class icon labeled with three stars ("* * *") is included in Figure 1 to indicate that there are other possible products.

The class, Constructed Past, is largely outside of the scope of this article, but it is worth noting that the framework for constructing the past, as illustrated in Figure 1, encompasses the concern of historiography to analyze instances of Constructed Past in light of the Inclinations and Intentional Domains that influenced them.

### 3.1.1. Sphere of Interest

This section further articulates the Sphere of Interest. Figure 2, Sphere of Interest, is a UML class diagram showing, as classes, the main things that may be taken into account given the time frame and Field of Interest defined in the Sphere of Interest. There are four such classes: Entity, Event, Process and State of Affairs. An Entity is something that existed. An Event is something that happened or was done. At least one Entity must be involved in every Event, but interest may be limited to an Entity apart from any involvement in an Event. A Process is a set of several related Events. State of Affairs is a configuration of one or more objects which have some characteristic(s) that are invariant during a period of time.

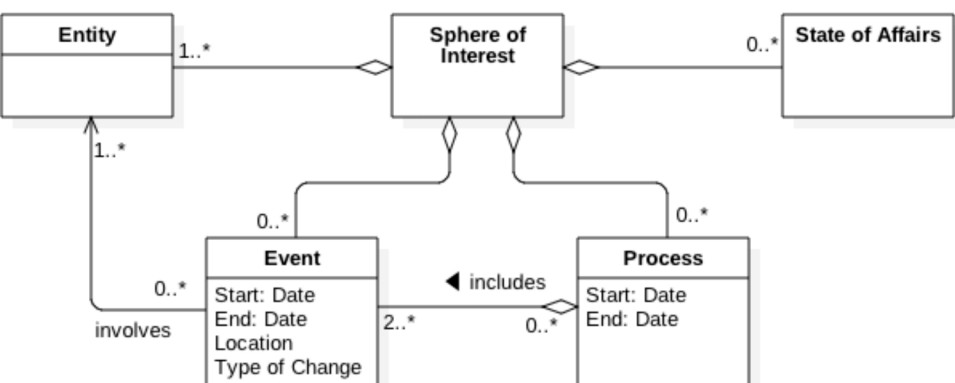

**Figure 2.** Sphere of Interest. The Sphere of Interest specifies what Entities, Events and States of Affairs are thought to be relevant in a Target Past. It may also include Processes, which are sets of related Events.

A Sphere of Interest can encompass many things in all four classes, but it must be about at least one thing. If only one, that object must be an Entity. Thus, the other possible elements in a Sphere of Interest have a multiplicity of zero or more. If the Target Past only aims to determine one or more properties of one or more Entities, the Sphere of Interest need not include any Event. If no Event is included, by definition, neither is any Process. A State of Affairs may focus on an Event or Process but, given that every Event must involve at least one Entity, the minimum of one Entity per Sphere

of Interest holds in this case. Similarly, a Sphere of Interest that only concerns characteristic(s) of one or more Entities, may not include any State of Affairs.

An Entity may or may not have a duration. Abstract concepts, such as time, justice, viscosity and monarchy, have no inherent temporal attribute, even though the instantiation, expression or understanding of such concepts may vary over time. Physical objects, such as organisms, buildings and electrical signals have durations. However, every Entity, whether conceptual or physical, must have at least one inherent property that is persistent. Persistent inherent properties of an object may be inherited from a class in which it belongs.

In an Event, something changes. What changes is one of the defining properties of an event. It will often be one or more Entities. However, an Event might alter a Process; for example, a power failure can interrupt the execution of a computer program. A second defining characteristic of an Event is the nature of the change. For example, enacting a law is different than applying the law in a judicial decision. Specifying when and where an Event occurred may also be necessary to identify the Event as the same thing may happen to an object multiple times. Every Event has a finite duration bounded by start and end times, at least one of which must be within the time frame of the Sphere of Interest.

Events and Processes, as well as their relationships with Entities, are further explored in Section 3.1.2. State of Affairs is addressed in more detail in Section 3.1.3.

### 3.1.2. Event, Process and Action

This section provides more details on the relationship between Event and Process and introduces a special type of Event, Action. Every Event involves at least one Entity. Figure 3, Entity Involvement in Event, illustrates the relationship between these two classes. The way an Entity is involved in an Event is specified in the class, Involvement. Involvement is an association class; that is, a class each instance of which associates single instances of two other classes. Involvement has subclasses; that is, there are different ways an Entity can be involved in an Event. Four common ways are shown in Figure 3. An Entity may have more than one Involvement in an Event, but an Involvement is specific to a single Event. If the Event did not happen, there could be no Involvement. Further, if the Event did not happen, no Entity would have any Involvement in it. Similarly, if the Entity did not exist, it could not have an Involvement in any Event.

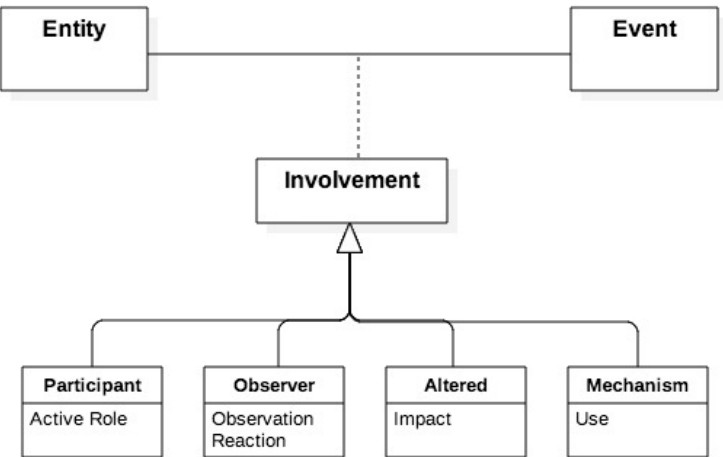

**Figure 3.** Entity Involvement in Event. Every Event must involve at least one Entity. An Entity may be involved in different ways, such as actively participating, observing, being impacted or being used.

The four subclasses of Involvement depicted in Figure 3 are Participant, Observer, Altered and Instrument. An Entity as a Participant has an Active Role in an Event, such as initiating, terminating, guiding or performing it. An Observer produces some Observation, which may or may not be persistent in the form in which it is originally made. An Observer might also have a Reaction to an Event and

the Reaction could be recorded in a persistent form. An Entity that is altered in an Event has some Impact that changes the Entity. An Impact is a direct consequence of the Event entailed by the nature of the change it involves. The Impact might be intentional; for example, when an environmental sensor transmits data it was designed to register. However, an Impact might be unintentional, such as civilian casualties in battle. An Instrument contributes to the initiation or completion of the Event. It might also be altered by the Event but that would be in the Altered role.

Figure 4, Event and Process, elucidates the relationship between Event and Process. An Event may occur without being part of a Process and, even if it is part of one, an Event may not be dependent on the completion of the Process. Hence, the relationship between Event and Process is one of aggregation, not composition and an Event might not be included in any Process. An extended Process may include one or more subprocesses.

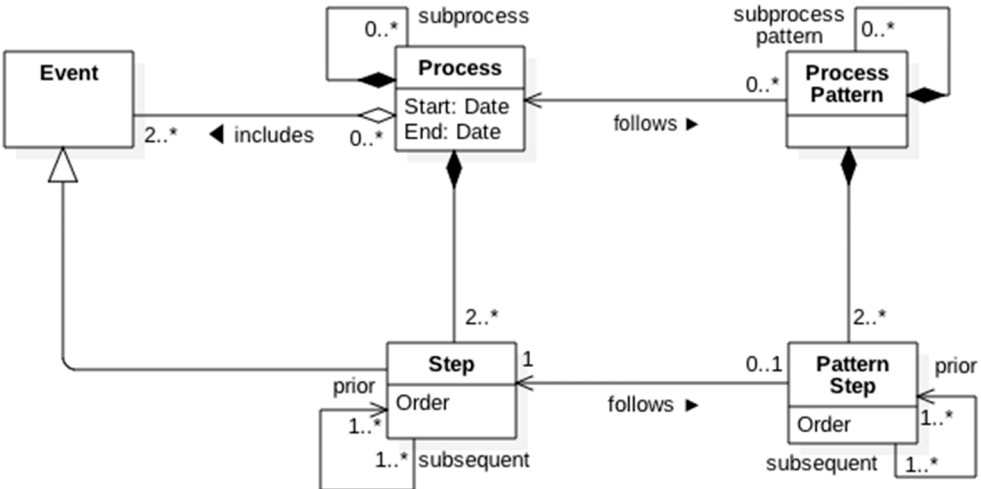

**Figure 4.** Event and Process. An Event may be part of a Process, which is a set of related Events. When it is, it is called a Step. A Process may follow a Process Pattern. When it does, the Steps in the actual Process may correspond to Pattern Steps in the Process Pattern; however, there may be actual steps in addition to those indicated in the Process Pattern.

A Process includes at least two Events as Steps, but may include many more. The subclass, Step, is defined as an Event which is part of a Process. Hence, a Step is dependent for its existence on Process and would cease to exist as a Step if the Process did not occur. Its relationship to Process is thus one of composition. Given that there must be several Steps in a Process, two Steps may be related as prior and subsequent. Steps may also be parallel to one another, but Figure 3 does not include this detail. The subclass will have at least one attribute not found in the superclass: Order or position in the Process.

A Process may follow a Process Pattern, which could be explicitly defined and imposed, for example, by regulation or corporate policy, or observed to occur habitually or regularly. The relationship is defined as "follows" rather than "adheres to" because an instance of a Process may conform by and large to a Process Pattern, but deviate from it in some respect. A complex Process could include subprocesses that may not follow a corresponding Process Pattern. The patterns that are followed by subprocesses in a Process may be subprocess patterns in an overarching Process Pattern, but subprocesses could also follow different, and even independent, Process Patterns. Hence, the relationship of Process Pattern to Process is zero or more.

A Process Pattern must include at least two Pattern Steps and there is a one-to-one correspondence between a Step and a Process Step which it follows. However, there may be Steps in a Process that do not strictly follow a Process Pattern. Even when a Process adheres to a Process Pattern, there may be Steps besides those in the Process Pattern. Hence, there may be zero or one Pattern Step for each Step in a Process.

Figure 5, Human Action, introduces a subclass of Event, Action, in which humans have an Active Role. Other subclasses of Event not addressed in this paper could be added to the model. Action is an Event in which at least one Human Agent is actively involved. Human Agent is a subclass of Participant. In addition to the Active Role that all Participants have, Human Agents characteristically participate in Events in a manner which furthers a particular Aim.

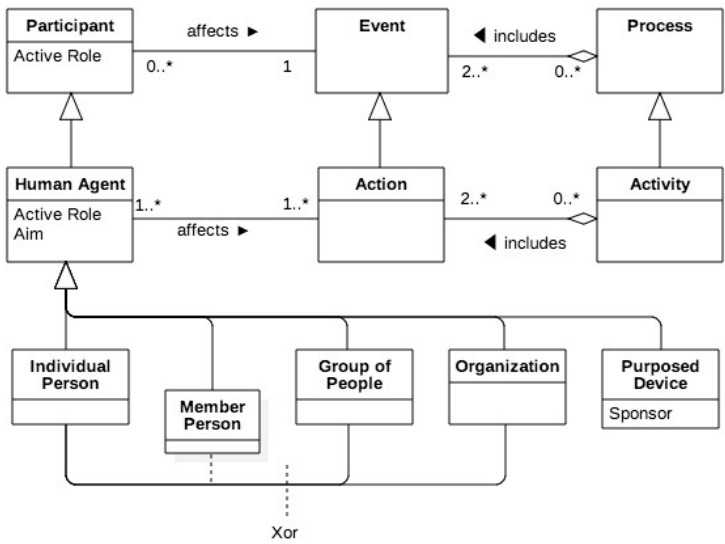

**Figure 5.** Human Action. When a Human Agent has an Active Role in an Event, the Event is specifically an instance of the subclass, Action. When the Action is part of a set of related Actions, the set is termed an Activity. Activity is a subclass of Process. Different kinds of Human Agent may be involved in an Action, as described in the text.

Five subclasses of Human Agent are distinguished: Individual Person, Member Person, Group of People, Organization or Purposed Device. An Individual Person is a biological human. A Group of People and Organizations include several individuals. The difference between a Group of People and an Organization is that a Group of People is defined by the Individual Persons who constitute the group and share a common purpose or interest, while an Organization has a structure and norms, and often a legal status, that transcends the set of Individual Persons who belong to it both at any given time and over time. Member Person is an association class that relates an Individual Person to either a Group of People or an Organization. Member Person acts in a role or capacity defined by a Group of People or Organization to which the individual belongs. These four subclasses are obvious types of Human Agent in the ordinary sense of that term, but Purposed Device is not intuitively a subclass of Human Agent. In CPT, a device qualifies as a Human Agent when it is put into place by an Individual Person, Member Person, Group of People or Organization to effect some purpose. Examples of Purposed Devices include computer applications, surveillance cameras, and traffic signals. An Action may involve no other Human Agent than a Purposed Device. For example, a fire may set off fire alarms that have the Active Role of warning and the Aim of helping people to avoid injury and minimize damage. Nevertheless, Purposed Device is an association class that relates a device to one of the other subclasses of Human Agent. Hence, Purposed Device is modeled with the attribute, Sponsor in Figure 5.

The subclasses of Human Agent are also subclasses of Entity. Instances of any of them may be objects in a Sphere of Interest without being involved in an Event as a Human Agent. For example, an Individual Person could be affected by an Event; a Group of People could observe an Event; an Organization could be formed by an Event; or a Purposed Device might be reverse engineered in an Event. Human Agents might also have instrumental, rather than intentional Involvement, in Events.

For example, human decisions in planning and actions in building a dam might be critical in its eventual collapse [34].

Just as a Process is a set of related Events, an Activity is a set of related Actions. The relationship between Human Agent and Action and between Action and Activity parallel those relating Participant, Event and Process. The details about Event and Process shown in Figure 4 are inherited by their subclasses, Action and Activity, respectively, as shown in Figure 5.

### 3.1.3. State of Affairs

The class, State of Affairs was introduced in Section 3.1.1 as one of the principal components of the Sphere of Interest. A State of Affairs is defined by a set of one or more assertions, all of which are true for the same chronological period and concern the same or related objects that are either instances of the Entity, Event or Relationship or their subclasses. Each assertion in the set is about either a single instance, or a single property of such an instance, or a single Relationship. When a Relationship characterizes a State of Affairs, the state includes the two related objects. The related instances may be of the same or different classes. If any of the defining assertions ceases to be true, the State of Affairs terminates. A Target Past could include several State of Affairs determined by different time spans, different sets of objects or different properties.

An assertion may be existential, qualitative or quantitative. It may stipulate whether an object existed or not for the duration of the State of Affairs. Alternatively, the State of Affairs might be determined based on whether an Event or Process endured throughout or did not occur at all during that time. For example, a period of peace is a time when there are no armed hostilities between two parties. An assertion about a State of Affairs may specify a qualitative or quantitative value of a property. The value need not be static; that is, the value might be a constant rate of change or even a constant acceleration. Moreover, the value may be the result of a Boolean expression of arbitrary complexity. Multiple assertions may be combined in a single complex expression.

The value specified in an assertion must be invariant throughout the State of Affairs; however, the property of which the value is specified must itself be variable because, if a property is invariant, the object is stateless with respect to that property.

The following statement exemplifies a State of Affairs: Barbara McClintock pursued graduate studies in cytology and genetics at Cornell University from 1923 to 1927. The statement asserts the persistence of a Relationship, student, between an instance, Dr. McClintock, of the class, Member Person, to an Organization, Cornell University, during a four-year period and specifies a qualitative property of the Relationship; namely, that it concerned cytology and genetics.

A second case illustrates a complex State of Affairs described in a single, complex expression: During Algeria's war of independence (1954–1962), successive governments in the French Fourth Republic insisted it was a purely internal affair, not an international one. The assertion fixes the time frame of the State of Affairs, associating it with a Process, the Algerian War of Independence, and makes it dependent on a qualitative condition; namely, assertions by French governments about the scope of the process. The condition is a Boolean assertion and would fail if it were shown that any government in the Fourth Republic held a different position during the war.

In the first instance given above, if the Purpose were to determine the role McClintock's graduate studies had on her eventual winning of the Nobel Prize, the Sphere of Interest might include things like the frequency of women in various roles (student, graduate assistant, professor) in science education in that time frame, the experience and careers of other women with similar education and careers and even broader subjects such as the state of the science during McClintock's student and professional years [35].

Similarly, if in the second case cited the Purpose were to discover whether the public statements of the Fourth Republic were consistent with its internal actions or dynamics, the Sphere of Interest would be expanded considerably, including exploring the impact of tactics used by the Algerian rebels

and examining the relations of France with other nations, especially the United States, and exploring the perceived self-interests and objectives of those nations [36,37].

The definition of a State of Affairs entails the inclusion of certain Entities, Events, Processes and Relationships in the Sphere of Interest; however, the Intent of Construction would be determinative of scope. If the Purpose of construction were to decide the truth or falsity of the assertions in the State of Affairs, the primary objects of interest would be the Entities, Events, Processes and relationships that are the subjects of these assertions. This would also be the case if the Purpose were to develop more detailed data about the State of Affairs. However, if the Purpose were to elucidate the impact of the State of Affairs, the contents of the Sphere of Interest could be expanded or reduced. If, for example, the Target Past focused on how McClintock earned the Nobel Prize, her graduate years might be of minor interest [38]. A State of Affairs could extend beyond the time frame of the Sphere of Interest. Alternatively, a State of Affairs might start or end within the time frame of the Sphere of Interest.

A State of Affairs, depicted in Figure 6, depends on the existence of one or more Persistent Properties or one or more Persistent Relationships or some combination of both. In any case, persistence means that a property or relationship has a Start Date and an End Date that defines a period of time which is at least partly within the time frame of the Sphere of Interest. If any Persistent Property or Persistent Relationship that characterizes a State of Affairs changed or went out of existence, the State of Affairs would end. Thus, the duration of a State of Affairs is the shortest intersection of the duration of a Persistent Property or Relationship with the time frame of the State of Affairs.

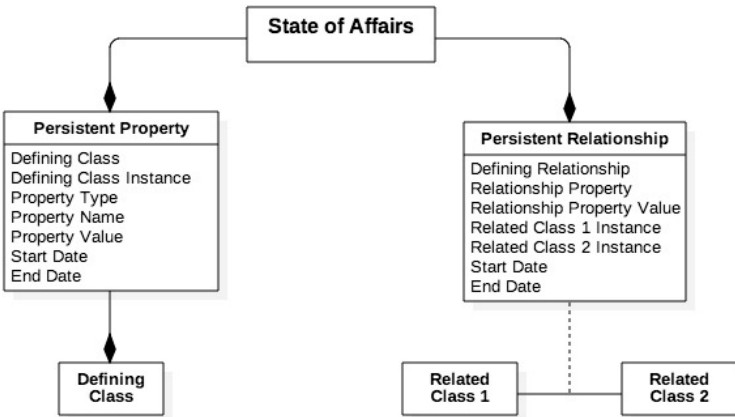

**Figure 6.** State of Affairs Internal Composition. A State of Affairs is determined by one or more Persistent Properties or Persistent Relationships. The relevant properties or relationships belong to one or more classes that exist within the Target Past.

A Persistent Property is an attribute or behavior of an instance of a Defining Class, which must be a class of Entity or Event that is part of the Target Past. An instance of Persistent Property identifies the Defining Class and the instance of that class that has the Persistent Property indicates what type of property and its name, and specifies the value of that property during the duration of the State of Affairs. The relationship between Persistent Property and Defining Class is one of composition because if Property Value changed or the identified instance of the Defining Class disappeared, so would the Persistent Property and thus the State of Affairs would be terminated.

In the McClintock example, as described above, a new class, University, would be needed and Student should be added as a subclass of Relationship and associated with both Individual Person and University. Student would need two persistent attributes in this State of Affairs. One, which might be named Student Status, would have enumerated values of undergraduate, masters and doctoral. The specifications for the State of Affairs would be satisfied if the actual value of the Barbara McClintock instance were either masters or doctoral. The other attribute could be named Area of Study, with possible values including all the major and minor fields offered by the Cornell College of Agriculture where she was enrolled. The actual values in her case would be cytology and genetics. In practice, the domain

model would need to be refined further to suit the case, including adding more specific classes of Entities such as University, School, Department, Scientific Discipline, etc., and more specific classes of Processes and Process Patterns for masters and doctoral studies.

To illustrate the need for additional specification of the model, consider the State of Affairs of the Fourth Republic. Central to this state is the relationship between successive governments and statements about the relationship of France to Algeria. The governments fall within the scope of Organizations as Human Agents. Statements accessible for construction of the Target Past are documents. To address the empirical situation, subclasses of Human Agents need to be added to represent not only government, but also government agencies and officials. Document must be defined as a subclass of Entity and a variety of subclasses of Document are needed. These examples only begin to illustrate the extensions that would be required to apply the model to this case.

The discussion of the McClintock and Fourth Republic cases should not be taken as indicating that the construction of instances of Target Past need to articulate a detailed model of a State of Affairs and populate it with related data. Rather, it is intended primarily to demonstrate that concepts in the CPT model are applicable empirically. The variety of possible constructions of the past is sufficient that there are undoubtedly some situations where articulating a more detailed formal model and organizing data within its structure would be advantageous. In other situations, the model might be applied heuristically to identify the objects and relationships that should be considered in pursuing a Target Past without being substantially elaborated.

### 3.2. Construction Materials

The concepts set out in the last section characterize the motivation, approach, scope and contents of past constructions. This section explores and describes the things that may be used in the construction of the past, as it were, its Construction Materials. Construction Material is a class whose instances contain or convey data useful in constructing a Target Past. Figure 7, Construction Materials in Context, is a class diagram that indicates the relationship of Construction Material to the framework of construction. Construction Material must relate to the Sphere of Interest either by providing context or data related to one or more Entities or Events within the sphere. To some degree, an instance of Construction Material should satisfy the Intent of Construction, such as by contributing to answering one or more questions, contributing to the content or structure of the Expected Outcome or satisfying the Purpose of the construction. It is likely to be used at some stage in an In-Progress Construction, although it may subsequently be eliminated. The most valuable Construction Materials are those that contribute to the Constructed Past, even if not explicitly referenced therein.

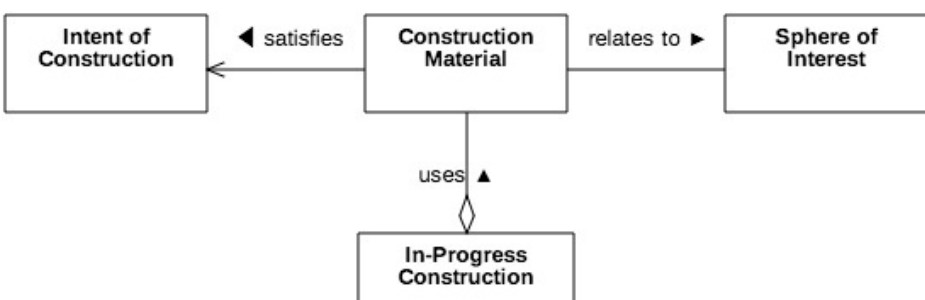

**Figure 7.** Construction Material in Context. Construction Material comprises information objects that are deemed relevant and potentially useful in constructing the past. Thus, they must relate to the Sphere of Interest and should satisfy the Intent of Construction. Relevant Construction Material is used in In-Progress Construction.

Thus, the selection and use of Construction Material is critical in producing the Constructed Past. Selection in this context means the determination that an item will be exploited in construction by extracting data from it; accepting assertions it makes, analyzing its contents, etc. The first criterion

of selection is that the item relates to the Sphere of Interest. The item should also be perceived as serving the Intent of Construction. Evaluation according to both criteria may be weighted according to how well it seems to serve the intent; to what extent it is congruent with the Sphere of Interest; how well it relates to key objects or events in the Sphere of Interest; and its potential for yielding new or improved data or understanding.

Initial selection may be heavily influenced by the Intent of Construction. Someone who wants to quickly gain familiarity with, or some depth of understanding of a Target Past would likely choose expert sources, such as audit reports, scholarly publications or encyclopedia entries. In contrast, someone who wants to develop original insights would prefer writings by Individual Persons within the Sphere of Interest. Selection could change by both additions and deletions as knowledge is gained in the process.

Construction Materials are either Vestiges or Reflections, as indicated in Figure 8, Construction Materials. A Vestige is an object that existed within and survives from the time frame of the Sphere of Interest. A Reflection is an information object produced in the course of construction. A Reflection typically expresses a cognitive reaction to one or more Vestiges or earlier Reflections. A Reflection might reflect an existing Constructed Past deemed relevant to the Intentional Domain.

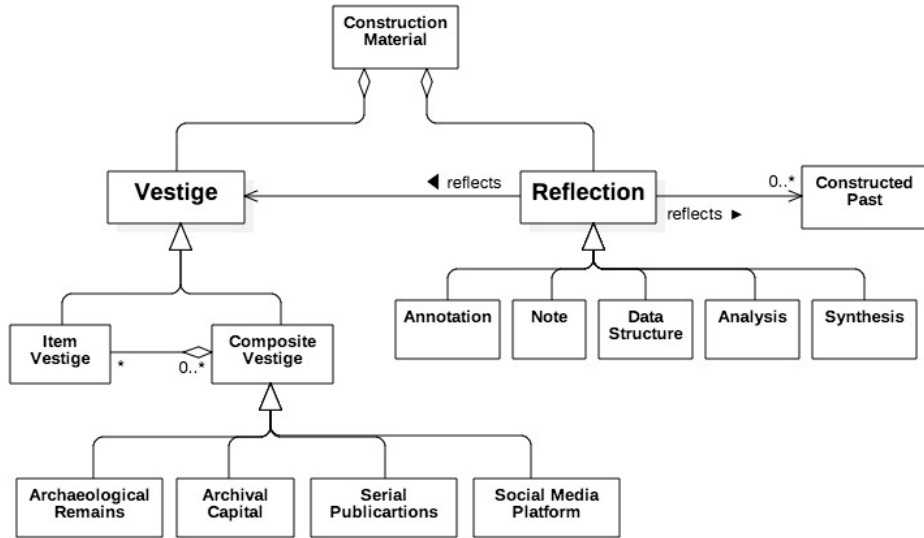

**Figure 8.** Construction Materials. The two main types of materials used in construction of the past are Vestiges and Reflections, shown as classes in the diagram. A Vestige is an object that existed in and survives from the time frame of the Sphere of Interest, while a Reflection is an information object created in the process of construction.

Both Vestige and Reflection have subclasses. The subclasses of Vestige are Item Vestige and Composite Vestige. An Item Vestige may be anything that satisfies the definition of Vestige. An Item Vestige may or may not be part of a Composite Vestige. Hence, Figure 8 showed Item Vestige as part of zero or more Composite Vestiges. A Composite Vestige is an aggregate of Item Vestiges that existed within the time frame of the Sphere of Interest. Figure 8 does not show it, but a Composite Vestige may be either ordered or unordered. The order, if any, must have existed within the time frame of the Sphere of Interest. It is possible that both the membership and the relationships within a Composite Vestige varied during that time frame. A Vestige Item may be discovered in isolation and subsequently associated with a Composite Vestige. Likewise, the ordering within a Composite Vestige might be determined progressively during the process of construction. However, all assertions about both the membership and relationships in a Composite Vestige must be based on evidence that is itself vestigial and sufficient to justify an estimate of a high probability that the composite existed within the specified time.

There are many possible types of Composite Vestige. Figure 8 shows four subclasses: Archaeological Remains, Archival Capital, Serial Publications and Social Media Platform. These four relate to a broad scope of time and illustrate different types of contemporaneous relationships. Social Media Platforms are recent phenomena. All items in a platform date from a brief period of time. Relationships on a platform are created ad hoc by users based on interests in given topics or themes [39]. Although the earliest serial record keeping dates back millennia [40], Serial Publications are a phenomenon of the modern era, even though they have existed for centuries longer than social media [41] (pp. 460–462). The Relationships of instances of Serial Publications are determined top-down according to a priori criteria [42].

Archival Capital is an adaptation of one of the principal concepts of archival science, that of the archival fonds. This concept was first developed in France and defined as "Un fonds d'archives est en effect l'ensemble des pièces de toute nature que tout corps administratif, toute personne physique or morale, a automatiquement et organiquement réuni en raison même de ses fonctions or d son activité" [43] (pp. 22–23). This may be rendered in English as the totality of information items of any type whatever that any administrative body or any physical or legal person automatically and organically assembled as a direct consequence of its functions or activities. The unfortunate translation of 'fonds d'archives' into 'archival fonds' loses the connotation it has in French of something of value, in this case, an ensemble of information assets. 'Archival assets' restores this connotation and has the additional benefit of avoiding the reduction that often occurs in practice of an archival fonds to the information assets managed in a formal record keeping regime. Frequently, and especially in the digital realm, even extensive sets of information assets are not subject to records management. This discussion leads to the recognition that there can be at least two types of Relationships within Archival Assets: those that result directly from the performance of functions and activities and those imposed in accordance with a filing system. The former are the relationships described in the concept of the archival bond [44].

A basic question, which will be addressed further in Section 4, is: what is an item in Archival Assets. The obvious answer that would be given by archivists and records managers is that it is a record. A record is "A document made or received in the course of a practical activity as an instrument or a by-product of such activity, and set aside for action or reference" [45]. Unfortunately, in practice, record is often treated as equivalent to document, ignoring that, by definition, a record is a document within a specific context. Instead of 'record', CPT adopts the German term of 'Archivalieneinheit'; that is, an Archival Unit, which is defined as a unit within a fonds [46]. This construct is both consistent with the concept of fonds d'archives and does not entail that an item in an instance of Archival Assets has been categorized or set aside as a record.

The basic composite in Archaeological Remains is an aggregate of archaeological materials that were all found in the same site and are related chronologically. Archaeological materials include artifacts, ecofacts, structures, and features associated with human activity [47]. Archaeological research can be described as a multi-layered construction of a Target Past. First, field work involves constructing a past that relates the items found in a site to one another, in effect taking the individual items discovered and defining them as members of a Contemporaneous Composite. This composite has two dimensions: the matrix which characterizes the site as a whole, and the provenience which specifies the position of an item within the matrix. The initial process also entails inferring the time frame of the composite from properties of the items in it. A second phase relates this composite to other Archeological Remains. In archaeological terms, this type of Constructed Past is characterized as an association of the Archeological Remains at different sites. The next phase involves analyzing how people affected the Archaeological Remains at a site. This enables constructing a past that infers things about the material culture, way of life, activities and even belief systems of the people who left the Archaeological Remains behind [48].

The subclasses of Reflection identified in Figure 8 are arranged from left to right roughly according to their relation to other instances of Construction Material. An Annotation is a Reflection that is

specifically linked, physically or conceptually, to one such instance. The related material may be either a Vestige or another Reflection. A Note is data about some aspect of the Intentional Domain created during the process of construction. Like an Annotation, a Note might be about another instance of Construction Material but is not tightly bound to it and it might be about several other instances. Furthermore, a Note might have a more general scope or be related to the process of construction rather than its contents. A Data Structure is a conceptual schema that defines categories of data within the Intentional Domain, as well as relationships between categories and among data objects. A Data Structure is used to organize Construction Materials or data extracted from them. A Data Structure is likely to reflect the Purpose, Expected Outcome and Questions identified in the Intent of Construction. As indicated by its name, Analysis is something that is produced by analyzing data collected or created in the course of construction. Similarly, Synthesis brings together data from different sources. While the Data Structure provides a way of assembling data, a Synthesis is an object produced by combining multiple pieces of data into a coherent whole. Its scope may extend to the entire Sphere of Interest, but it might be limited to only part of it. A Synthesis may be organized in a way that is parallel to the Data Structure, but it might also be shaped by one or more Questions in the Intent of Construction, including issues that arise in the course of construction. Additional subclasses of Reflection not identified in Figure 8 may be added to the model. Data Structure, Analysis, and Synthesis might be used to qualify an item as vestigial or as the basis for asserting the existence, membership and structure of a Composite Vestige. They might also be used to define composites that are artifacts of the construction, rather than vestigial.

## 4. Towards Testing, Verification and Quantification

The presentation of CPT in Part 3 is highly abstract and requires the addition of more specific classes and properties, with greater precision, to be applicable in practice; however, between articulation and application, the critical issue that needs to be addressed is whether CPT can support empirical testing and verification. To enable the reformulation of archival science as an engineering discipline, verification should be against quantitative parameters. Reformulating qualitative concepts in quantitative terms has the further benefit of creating the opportunity to apply a variety of powerful and supple mathematical tools in the construction of the past.

In the development of CPT, the issue of verifiability has been addressed from the bottom-up, starting with Construction Materials and focusing on Archival Assets. An Item Vestige that is a member of an instance of Archival Assets may be called an Archival Vestige. A record, as defined in archival science [49], is a subclass of Archival Vestige. Archival Assets that comprise records include files, series and archival fonds, progressively higher levels of aggregation of records in accordance with a filing system or records classification system [50]. The systematic arrangement of records in such systems typically has a tree structure. A tree is a type of graph, suggesting the possibility of adopting a graph theoretical approach in archival engineering.

This possibility becomes compelling in light of the fundamental concept of archival science. "At the core of archival science is the concept of the archival bond, that is, the network of relationships that each record has with the records belonging in the same aggregation. The archival bond first arises when a record is set aside and thereby connected to another in the course of action, but is incremental, because, as the connective tissue that joins a record to those surrounding it, it is in continuing formation and growth until the aggregation in which the record belongs is no longer subject to expansion, that is, until the activity producing such aggregation is completed" [51]. There are several problems with the way this concept has been articulated. Obviously it deviates from Cencetti's original formulation where the archival bond arises from the use of documents by the same Human Agent in the same Activity, thus prior to and independently of being set aside as a record. Moreover, neither the quoted description nor Cencetti's formulation distinguish between the network as a whole and the specific relationships between pairs or among groups of records. Additionally, the application of the concept of archival bond has assumed that all the relationships that arise in use are embodied in the filing

system in which the records are kept [52]. Except in very simple cases, a filing system is a partial way of expressing relationships among records; partial in both senses of incomplete and biased. By locating a record in a single position in a hierarchical classification, filing systems constrain the expression of the relationships of a document to other records and to the action or actions in which it was used [8].

These confusions and other difficulties can be eliminated by introducing the concept of an archival graph. An archival graph is a graph whose nodes are the Archival Units used in an activity and whose edges are the Relationships that arise from that use. Strictly speaking, in conformance with graph theory, an Archival Unit is itself a graph whose nodes are an Item Vestige and the Relationships it has with other Item Vestiges used in the same Activity. It is a directed graph where each edge goes from an Item Vestige to a Relationship with a "has a" label. This construct eliminates the confusion entailed by the fact that the same document, or information item of any kind can be a different record in different contexts. For example, an invoice identifies an account receivable in the records of a supplier, but an account payable as a record of the customer.

The definition of an archival graph encompasses a domain that is larger than that of the archival bond when there is more than one independent Human Agent participating in an activity. That is because 'record' is defined with respect to a single Individual Person, Group of Persons or Organization acting as a records creator. If several independent Human Agents participate in an Activity, each may keep its own records. While some information objects will likely be duplicated in the different aggregates, just as likely there will be records unique to each. Furthermore, the Archival Units used in many activities may include some that are not kept as records by any Participant; for example, Human Agents may have frequent recourse to data available on the Internet and not downloaded to any local store [53].

In addition to expanding the domain, representing Archival Assets as a graph enables greater clarity in the application of established archival concepts. For example, ignoring the troublesome equation of the archival bond with a record-keeping system, the archival bond can be conceptualized as a subgraph of the total graph of all Archival Units used in a given activity where the subgraph is induced by selecting those vertices accumulated by an Organization, Group of Persons or Individual Person performing an Activity in an independent capacity. Conversely, combining the subgraphs of all Archival Units accumulated in all the Activities of an independent Human Agent would produce a map of the archival fonds of that agent. Furthermore, the union of the graph of the archival fonds with the graph of the agent's record-keeping system could yield significant insights into the discrepancies between how the agent actually carried out its activities and how it identified and managed information assets that it deemed to have persistent value.

While an archival graph, like archival fonds, is limited to the Archival Assets of a single Human Agent, archival graphs of all the agents involved in an Activity can be combined, a significant advantage in any Target Past that encompasses the whole of an Activity in which multiple parties have an Involvement.

A graph theoretical approach can also help to resolve divisive issues in the archival field. Records and records aggregates have traditionally been limited to the records of a single records creator. However, there are cases where similar records are created and maintained in a single aggregate maintained by different, successive records creators. This is fairly common in large organizations, such as governments, government agencies and large corporations, which repeatedly reorganize over time. In such cases, successive records creators many not only create similar records and record aggregates, but they may also inherit assets of their predecessors [54]. While each instance of Archival Assets is associated with a single Human Agent, a graph can be defined to encompass all the records and record aggregates accumulated in the exercise of a specified function. This graph would be the intersection of the subgraphs of all Archival Assets containing records that meet the defining characteristics.

Archival graphs could also contribute to the construction of the past in combination with graphs of other objects, such as the subjects described in Archival Units, parties involved in or affected by Activities, and conditions (e.g., laws, documentary forms, resources) that affected Activities. While the

Records-in-Contexts model and ontology sponsored by the International Council on Archives is still in development, a beta implementation by the National Archives of France shows how this can be done. In addition to the usual, hierarchical description of a government agency, in this case la Direction générale des arts et des lettres, this implementation includes chronological links to predecessor and successor organizations, and associations with other organizations and individuals [55]. Such a combined graph, which could be termed an Archival Context Graph, would constitute a Data Structure in a construction of the past. Analysis of the Data Structure might be elaborated in an In-Progress Construction. These examples illustrate how articulation of formal models and reformulation of existing concepts applicable to the essential archival objectives of preserving and providing access to vestiges of the past in graph theoretical terms can contribute to the construction of the past.

This approach could also provide data and insights on ways to improve the operations of archival institutions; for example, by bridging the endemic divide between back-end and front-end functions in archival practice. Back-end activities are those that result in Archival Assets being preserved, while front-end activities enable the discovery by researchers of Archival Assets that are relevant to their interests and the delivery to them of both assets and information resources that enable understanding and interpretation of the data conveyed by the Archival Assets. In essence, the determination of what records should be preserved for the long-term is independent of any consideration of the potential value of the records to later users. It is a retrospective appraisal of what records constitute an adequate and authentic representation of the Entities and Events one wishes to document. This is not unreasonable given that there is a high degree of uncertainty about the interests of future users and that many uses of preserved records are orthogonal to the purposes for which the records were created and kept originally. The retrospective orientation of appraisal is echoed in the way records are preserved. Long-standing principles guiding archival preservation are that preservation should respect both provenance, where the records came from, and original order, how the records creator organized its records [41,56,57]. These principles recognize that a single, common criterion that determines the membership and structure of the archival bond is the Human Agent responsible for the records and that this datum and the relationships embodied in the tree implemented in a record-keeping system illuminate the contemporaneous context of the Entities and Events represented in the records. However, the interests of researchers engaged in construction of Target Pasts may well span several records creators and, even within the Archival Assets of a single creator, may not map to the order imposed on the records by the creator [58]. Abandoning these principles would result in the loss of important contextual data. Furthermore, besides being impractical, reorganizing records to suit the interests of a researcher would disadvantage others with different interests.

However, if an archival institution stored data identifying records creators, records and record aggregates in a graph-oriented database, the graph of all the holdings of the archives would constitute a universal set. A researcher could define and progressively build a graph corresponding to its Sphere of Interest. The intersection of a researcher graph with the universal set would map the researcher's interests to relevant members of the universal set, regardless of the boundaries of provenance and original order. This intersection would not impede navigating from members of the intersection to related contextual data. The institution could analyze these intersections to better understand the use of its holdings and consequently improve its front-end services. Accumulation of data about the intersections might also help the archives to evaluate whether its appraisal criteria led to adequate, or alternatively, excessive, representations of the targeted Entities and Events.

## 5. Overview

From the perspective of pragmatic information theory, being informed about the past occurs on receipt of a Constructed Past. The receipt may be via a persistent object, such as a book, transitory signals, such as the display of a view on a database on a video device, by oral report from a person or other vehicle. The introduction sketched an epistemology that informs the Constructed Past Theory that is elaborated in Part 3. According to this epistemology, (1) data exist as signs or are delivered as

signals in contexts that enable their interpretation; (2) information is an event wherein a set of signals is received, producing either a behavior or a change of state in the recipient; and (3) to know is to utter a proposition, where a proposition is an expression that has an objective meaning that can assigned some truth value or credibility. In terms of knowing the past, when a recipient is informed about the past, the result is either the behavior of uttering one or more propositions about the past or a change of state that consists of storing data about the past in a manner that enables such propositions to be formulated thereafter. The storage could be in personal memory, in solid state or magnetic storage media, in written text or other means.

The Constructed Past has informative value to the extent that it satisfies an intellectual or pragmatic objective. In its articulation of a framework for construction of the past and in concept of the Intentional Domain in particular, Constructed Past Theory relates the objective of being informed about the past not only to the Constructed Past but also to the process and materials of its construction.

In knowing the past, context is critical both in the epistemological sense just described and also in the process and output of producing a construction. Inclination and Intentional Domain form the context in which construction of the past is undertaken and brought to completion. Initial articulation of the Intent of Construction and Sphere of Interest enable assessment of whether the Target Past will satisfy the intended Purpose for the construction. Application of the Intentional Domain in the development of In-Progress Construction both guides the process toward achievement of its Purpose and facilitates improvements in the Expected Outcome.

This iterative relationship also guides the selection, processing and use of Construction Materials. CPT defines a catalogue raisonné of Construction Materials, characterizing the differences between Existing Constructed Pasts, Reflections and Contemporaneous Materials as date sources. This characterization can be used in appraising not only how well Construction Materials serve the Purpose of the construction but also the extent to which they are capable of supporting the Expected Outcome given the Level of Effort defined in the Intent of Construction. CPT provides a basis for optimizing the values of objectivity and originality in constructing the past through its delineation of Contemporaneous Materials and especially, in clarifying how different kinds of Contemporaneous Complexes enable recovering original context; that is, the sets of relationships among objects that existed within the Sphere of Interest.

This paper presents a germinal articulation of Constructed Past Theory rather than a definitive formulation. It attempts to demonstrate how the approach taken in CPT can represent various and arbitrarily complex past situations, as well as different motivations for investigating the past and different uses for the results. Nevertheless, the past is undoubtedly more complex and varied than represented in the diagrams. Additional classes, relationships and operations can and should be added and integrated within CPT.

**Funding:** This research received no external funding.

**Acknowledgments:** I thank Luciana Duranti for critical insights into archival science, Daryll Prescott for advice on the use of UML, and Vicki Lemieux, Richard Marciano and William Underwood for encouraging this research.

**Conflicts of Interest:** The author declares no conflict of interest.

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
