# Peer review of "The Construction of the Past: Towards a Theory for Knowing the Past"

_information, doi:10.3390/info10110332_

Round 1

Reviewer 1 Report

In this article, the author introduces Constructed Past Theory (CPT), whose aim is to support implementation in automated systems.

The author claims that "the theory does not accept the distinction between data and information that is common in information technology" and it provides some examples. I find this an interesting point, which is, I believe, central to this work.

Furthermore, "Data are structured or, more accurately, exist within structures that enable their interpretation". This is also very interesting, which would have some useful applications in other contexts, such as Topological Data Science and general mathematical formulation of data. Therefore, I would suggest looking into these fields to explore potential overlapping and applications.

Figures are helpful and well designed but I would perhaps include some more information in their captions.

Overall, this is a very well written and interesting manuscript.

Author Response

Thank you for your insightful comments.

In response to your check box suggestions for improvement, I have added information in the introduction about the research behind this article and I have made numerous changes in the text and diagrams in the interest of greater clarify.

I have also added descriptive information to the diagram captions.

I added a new section 4 addressing the formulation of data in graph theoretical terms in response to your comment about math.

Reviewer 2 Report

This is a well-written and clearly-illustrated paper that deserves publication.

There is, however, an obvious question that is not addressed: how does CPT relate to theories of the construction of the past in human memories? It is now well-established that human memory, particularly episodic memory, is constructed. It is equally well-established that retrieving, using, and reconsolidating an episodic memory alters its contents. The language of entities, events, processes, goals, etc. is equally employed in models of human memory construction; see e.g. C. Fields, "The very same thing: Extending the object token concept to incorporate causal constraints on individual identity" Adv. Cogn. Psychol. 8 (2012) 234-247 and references therein. A brief discussion of the similarities - which seem quite striking - between CPT as an engineering theory and empirically-motivated models of the mechanisms by which episodic memories are constructed might be a nice addition to the paper.

The issue of episodic memory reconsolidation (e.g. L. Schwabe et al., "Reconsolidation of Human Memory: Brain Mechanisms and Clinical Relevance" Biol. Psychiatry 76 (2014) 274-280) raises the question of whether, and if so how, use of an archive alters the content of the archive, and whether meta-archives tracking such use and the consequences of such use form a class of particular interest within CPT.

Some minor issues:

p. 5: The phrase "Target Past" appearing above the caption of Fig. 1 should be part of the previous paragraph.

p. 9: "Other subclasses of Event not addressed in this paperboy be added to the model." Spellcheck error.

p. 10: "Five subclasses of Human Agent are distinguished: Individual Person, Group of People,
Organization or Purposed Device." There are only four. Following sentence needs correction also.

p. 15: "Social Medial Platform" Spellcheck error

p. 16: "Constructed Past Theory as presented in this paper is rather than definitive." Is what rather than definitive?

Author Response

Thank you for your helpful comments.

I have corrected each of the errors you pointed out on pp. 5, 9, 15 and 16.
Regarding the problem with subclasses of Human Agent on p. 10, there should be five subclasses. I have revised the text and replaced the diagram to include the fifth subclass, Member Person.

The question you identify regarding the relationship between CPT and human memory is stimulating. There are obvious parallels in the way the formation of episodic memory is modeled and CPT. However, this is not an area where I have any competency. Therefore, addressing the issue in an addition to the paper would at best be superficial.

Moreover there are substantial empirical and conceptual issues in the relationship between CPT and memory formation. Empirically, in contrast to the “same thing” phenomena in human memory, those engaged in construction of the past have no direct access to either the objects of their attention or the contexts in which those objects were bound. Access is limited to vestiges and imprints which are typically incomplete and slanted. Furthermore, many efforts to construct the past involve things with which the constructor has no prior contact or awareness.

Conceptually, the products of constructing the past would seem to be more appropriately categorized as semantic memory, rather than episodic memory.

Nevertheless, given the obvious parallels between the two domains, this is an area I would be interested in exploring if I could find a collaborator with appropriate expertise.